# Osteoporosis and Fracture Risk in Ovarian Cancer: Beyond the Oncologic Burden

**DOI:** 10.3390/diagnostics15232966

**Published:** 2025-11-22

**Authors:** Mariagrazia Irene Mineo, Giorgio Arnaldi, Valentina Guarnotta

**Affiliations:** Department of Health Promotion, Mother and Child Care, Internal Medicine and Medical Specialties “G. D’Alessandro” (PROMISE), Section of Endocrinology, University of Palermo, Piazza delle Cliniche 2, 90127 Palermo, Italy; mariagraziairenemineo@gmail.com (M.I.M.); giorgio.arnaldi@unipa.it (G.A.)

**Keywords:** bone loss, fractures, chemotherapy, bisphosphonates

## Abstract

**Background/Objectives**: Osteoporosis is a prevalent condition characterized by reduced bone mass and microarchitectural deterioration, resulting in increased fracture risk. Ovarian cancer represents a paradigmatic model of cancer-related bone loss, owing to the combined effects of abrupt surgical menopause, chemotherapy, and tumor-driven pro-resorptive mechanisms. **Methods**: We conducted a narrative review of the literature on skeletal health in ovarian cancer. We synthesized current evidence on the pathogenesis, diagnostic strategies, and management of bone loss in women with ovarian cancer, with the aim of providing a disease-specific framework for clinical practice. **Results**: Available evidence highlights a multifactorial “triple hit” to bone health in ovarian cancer: accelerated estrogen deficiency following bilateral salpingo-oophorectomy, direct tumor-derived stimulation of osteoclast activity, and chemotherapy-induced skeletal toxicity. Despite the high incidence of bone loss and fractures, systematic bone health assessment is rarely integrated into oncological care. Dual-energy X-ray absorptiometry (DXA) remains the cornerstone diagnostic tool, complemented by vertebral morphometry and fracture risk algorithms. Antiresorptive therapies, particularly bisphosphonates and denosumab, together with calcium, vitamin D, exercise, and fall-prevention strategies, have demonstrated efficacy in reducing fracture risk, although disease-specific guidelines are still lacking. **Conclusions**: Fracture prevention in ovarian cancer survivors is often overlooked despite its significant impact on morbidity and quality of life. Integrating bone health assessment and early antiresorptive therapy into care pathways is warranted, and future studies should develop tailored guidelines to make bone protection a key element of survivorship care.

## 1. Introduction

Osteoporosis is a widespread condition resulting from both quantitative and qualitative alterations of bone tissue. It is characterized by a reduction in bone mass and deterioration of bone microarchitecture, leading to an increased risk of fragility fractures. The global prevalence of osteoporosis ranges from 19.8% to 26.9% in women and from 9.6% to 14.1% in men, reaching up to 39.5% in African populations. In the United States, up to 40% of women and 13% of men are expected to sustain a fragility fracture during their lifetime [1,2]. The disease affects both sexes and can occur as a primary condition, associated with progressive ageing or menopause, or as a secondary form, arising from various disorders that interfere with bone metabolism, including oncological diseases [3]. In oncological settings, skeletal health is compromised not only by baseline risk factors but also by the combined effects of tumor biology and systemic treatments. As a result, fracture incidence is higher in cancer survivors than in the general population and has a measurable impact on morbidity, functional independence, and survival [4,5]. The risk of fractures in cancer patients increases as early as one year after diagnosis and remains elevated for up to 10 years [4]. Indeed, in this population, the incidence of major fractures has been reported to be approximately 6% at a median follow-up of six years [6], and this condition also appears to negatively impact survival. In addition, the risk of falls is higher in cancer patients, further increasing the incidence of fragility fractures. This is driven by non-modifiable risk factors such as age, sex, tumor stage, severity, and location, as well as modifiable factors including physical and cognitive function, balance and gait, body mass index and nutritional status, polypharmacy, muscle strength, and mood [7]. Within this scenario, ovarian cancer represents a paradigmatic model of cancer-related bone loss. The disease and its treatment exert a “triple hit” on skeletal integrity. Abrupt hypoestrogenism after bilateral salpingo-oophorectomy accelerates osteoclast-mediated resorption far more rapidly than natural menopause. Chemotherapy and systemic therapies further disrupt bone turnover, often in parallel with sarcopenia and functional decline. Tumor-derived factors create a pro-resorptive and inflammatory milieu that exacerbates skeletal fragility beyond baseline demographic risk.

Compared with breast cancer, where the pathophysiology of bone loss and the role of antiresorptive therapy are well defined, data regarding skeletal health in ovarian cancer are limited and heterogeneous. Ovarian cancer represents a distinct model of cancer-induced bone fragility, in which abrupt estrogen deprivation, systemic inflammation, and cytotoxic therapies synergistically accelerate bone resorption.

Despite the considerable clinical burden, bone health remains an under-recognized and inconsistently addressed component of ovarian cancer management. The available evidence is dispersed across general osteoporosis research, oncology cohorts, and preclinical investigations, with disease-specific guidance still lacking. This is in contrast to other forms of cancer that affect women more frequently, such as breast cancer, for which there are more detailed and structured guidelines regarding bone health management. This review seeks to integrate current knowledge on the relationship between ovarian cancer and skeletal health, with particular emphasis on the mechanisms driving cancer-related bone loss, the strategies available for diagnosis and monitoring, and therapeutic interventions designed to mitigate fracture risk in this patient population, trying to outline a practical guide, currently unavailable, for the management of bone health in these patients.

## 2. Osteoporosis: Epidemiology and Diagnostic Tools

Osteoporosis is characterized by quantitative and qualitative alterations in bone tissue, with reduced bone mass and impaired microarchitecture, leading to increased fracture risk. It is caused by impaired bone turnover, with an imbalance between the anabolic activity of osteoblasts, responsible for bone formation and the catabolic activity of osteoclasts, which mediate bone resorption. The predominance of osteoclastic activity ultimately increases the risk of fracture.

Prevalence rates are higher in developing countries (20.1–24.1%) compared with developed countries (11.5–17.7%) [8]. Women are more frequently affected in the European Union, prevalence exceeds 22% in women versus 6.6% in men, and fragility fractures also occur more often in women [9]. Low bone mass density (BMD) is independently associated with increased mortality in both sexes [10], particularly in men after fragility fractures [11]. In Europe, fracture incidence is expected to reach 3.3 million by 2030, with healthcare costs projected to rise from €37.5 billion in 2017 by 27% [12]. This rising prevalence reflects both the progressive ageing of the population and the fact that primary osteoporosis is closely linked to menopause and ageing. In addition, secondary osteoporosis should also be considered, as it is associated with multiple pathological conditions, including endocrine, gastrointestinal, hematological, rheumatological, and oncological diseases [13], thus greatly expanding the population at risk. Several diagnostic tools are available, with dual-energy X-ray absorptiometry (DXA) considered the gold standard. DXA provides quantitative measurement of BMD and fracture risk expressed as a T-score. A T-score of ≤−2.5 standard deviations defines osteoporosis. However, DXA has limitations, including artefacts caused by calcifications, osteophytes, vertebral fractures, or metallic implants.

To complement DXA, software such as the Trabecular Bone Score (TBS) (v. 2.1 TBS iNsigh) has been developed to assess trabecular microarchitecture, providing both quantitative and qualitative insights into bone status. High TBS values higher than 1310 are associated with normal microarchitecture, while values lower than 1230 show degraded microarchitecture. These data are crucial for assessing fracture risk, particularly in secondary osteoporosis, where fracture risk does not always correlate with BMD or T-score values.

Emerging and complementary tools include radiofrequency echographic multi-spectrometry (REMS), magnetic resonance imaging (MRI), and computed tomography-based approaches. REMS allows rapid ultrasound-based evaluation of BMD at axial reference sites (lumbar spine and proximal femur), providing T-scores and Z-scores [14]. REMS demonstrates high reliability, with good concordance with DXA [15] and low intra- and inter-operator variability in predicting fracture risk using the Fragility Score [16]. Magnetic resonance imaging (MRI) can assess vertebral bone quality using T1-weighted sequences, distinguishing healthy from osteopenic/osteoporotic bone with an accuracy of 81% [17]. Computer tomography (CT)-based approaches allow separate evaluation of trabecular and cortical compartments. Quantitative CT (QCT) provides volumetric assessment of BMD in the lumbar spine and proximal femur, although only T-scores from the proximal femur and total hip are accepted for diagnostic purposes. High-resolution peripheral QCT (HR-pQCT) represents the most advanced tool for assessing bone quality, as it measures total, trabecular, and cortical compartments, and provides additional parameters such as cortical vBMD, trabecular thickness, and bone stiffness [18]. Other diagnostic tools include conventional lateral radiography for diagnosing and characterizing vertebral fractures, as it provides a visual assessment of the vertebral body shape and height loss. To standardize this assessment, the Genant Semi-Quantitative (SQ) scale is the most widely used tool, allowing clinicians to grade the severity of a fracture from Grade 1 (mild) to Grade 3 (severe) based on the percentage of height reduction, which is vital for clinical trials and predicting future fracture risk [19].

Clinical risk prediction can be refined using the fracture risk assessment tool (FRAX), which integrates several clinical variables, including BMD, to estimate 10-year fracture probability. FRAX recommends initiating active therapy when the 10-year risk of a major osteoporotic fracture is ≥20% or the 10-year risk of hip fracture is ≥3% [20].

## 3. Ovarian Cancer: Epidemiology and Clinical Context

Ovarian cancer ranks as the fifth leading cause of cancer-related death in women and remains the most lethal gynecological malignancy, with a 5-year survival of about 49% [21]. It is typically diagnosed at an advanced stage due to its often non-specific symptoms, and characterized by a high rate of recurrence following initial treatment [22]. Risk factors include reproductive and hormonal factors (nulliparity, anovulation, endometriosis), benign ovarian cysts, family or personal history of breast cancer, and inherited mutations such as BRCA1/2. In carriers, lifetime risk rises from ~10% at age 50 to ~49% by age 80, though preventive salpingo-oophorectomy substantially reduces incidence [23]. In this population, preventive salpingo-oophorectomy has been shown to significantly reduce the risk of developing ovarian cancer [24]. Ovarian cancer is broadly categorized into three groups: epithelial, germ cell, and sex cord-stromal tumors [25]. Epithelial ovarian cancer is by far the most common, accounting for approximately 95% of cases. Among epithelial tumors, the major histological subtypes are high-grade serous carcinoma (HGSC) (70–80%), borderline tumors (15%), and low-grade serous carcinoma (LGSC) (5–10%). Germ cell tumors represent 58% of ovarian tumors in women under 20 years of age, with the following distribution: dysgerminoma (33%), yolk sac tumor (14–20%), embryonal carcinoma (4%), non-gestational choriocarcinoma (2%), teratomas, and immature teratomas (36%). Sex cord-stromal tumors, which include subtypes such as thecoma, fibrosarcoma, Leydig cell tumor, steroid cell tumor, and both adult and juvenile granulosa cell tumors, account for around 8% of ovarian neoplasms [26]. Standard management involves cytoreductive surgery, typically hysterectomy with bilateral salpingo-oophorectomy, followed by platinum-based chemotherapy. Fertility-sparing approaches may be considered in selected women with stage I epithelial disease [27]. First-line adjuvant systemic therapy is based on platinum-based chemotherapy, with the possible addition of bevacizumab in advanced stages, and alternative regimens may be selected depending on tumor subtype [28]. While these therapeutic strategies have significantly improved survival, they impose substantial effects on bone health.

## 4. Mechanisms of Bone Loss in Ovarian Cancer

Radical surgery with bilateral salpingo-oophorectomy induces abrupt surgical menopause, with a rapid decline in circulating estrogen levels far exceeding that observed in natural menopause. This sudden estrogen deprivation leads to the immediate onset of typical menopausal symptoms, including vasomotor disturbances, night sweats, dyspareunia, and insomnia, along with alterations in bone metabolism [29,30]. The mechanisms involved in this process are reported in Table 1.

### 4.1. Hormonal Mechanisms

Normal bone metabolism depends on the balance between osteoclastic bone resorption and osteoblastic bone formation. Among the hormones that regulate these processes, estrogens play a central role.

Estrogens act on both osteoclasts and osteoblasts through binding to estrogen receptors, in particular ERα and ERβ. ERα is the predominant isoform in cortical bone, while ERβ is more abundant in trabecular bone; overall, ERα mediates most estrogen actions in bone cells [31]. Estrogens inhibit osteoclast activation primarily through the RANK/RANKL pathway by increasing the production of osteoprotegerin (OPG), which binds to and neutralizes RANKL. Normally, RANKL binds to its receptor RANK on osteoclast precursors and mature osteoclasts, stimulating their differentiation and activation, thereby promoting bone resorption [32]. Estrogens also induce osteoclast apoptosis via the Fas/FasL pathway [33] and reduce levels of pro-inflammatory cytokines, such as TNF and IL-1, which stimulate osteoclastogenesis [34]. Indeed, estrogen deficiency, whether following ovariectomy or during physiological menopause, has been shown to increase TNF production by T cells to levels sufficient to drive osteoclastogenesis [35]. In osteoblasts, estrogens promote survival by upregulating the anti-apoptotic protein Bcl-2, reducing oxidative stress [36], and mediating mechanotransduction processes by influencing the expression and function of mechanosensitive proteins [37]. Thus, estrogen deficiency negatively impacts bone metabolism, causing thinning and disruption of trabecular microarchitecture [31]. In women with ovarian cancer treated with radical surgery, the abrupt fall in circulating estrogen levels leads to a rapid reduction in BMD, evident as early as one year post-surgery. The extent of BMD loss is influenced by preoperative BMD, age, BMI, and type of treatment. In a cohort of 75 women after bilateral salpingo-oophorectomy, Lee JE et al. reported significantly lower lumbar spine BMD (L1–L4) at one year compared with controls. At follow-up, 23% of patients had normal BMD, 37% osteopenia, and 15% osteoporosis [38]. Similarly, Sobecki J et al. studied 185 women treated with salpingo-oophorectomy for gynecological tumors and observed a progressive and significant decline in BMD at 1, 3 and 5 years [39]. Predictive factors for osteoporosis included baseline BMD, prior chemotherapy, and the cumulative number of chemotherapy cycles. Comparable findings have been reported in BRCA1/2 mutation carriers undergoing prophylactic bilateral salpingo-oophorectomy. Garcia et al. reported that only 32% of women after surgical menopause maintained normal BMD after one year, while 55.6% developed osteopenia, 12.1% osteoporosis, and 4% experienced atraumatic fractures [40]. The detrimental impact of surgical menopause is particularly marked in women under 45 years of age, in whom differences in BMD compared with controls are more pronounced [41]. Similarly, a study conducted on a group of 469 women who underwent prophylactic salpigectomy showed that the impact of surgical menopause on BMD was much more evident in premenopausal women than in postmenopausal women at the time of surgery, with a significantly lower Z score being found in both the lumbar spine and the femoral neck at the time of the check-up [42]. However, not all studies are concordant. Large-scale analyses comparing premenopausal women undergoing salpingo-oophorectomy with women in natural menopause found no significantly higher incidence of non-vertebral fractures in the surgical group, even among those who had never used oral estrogens [41]. Similarly, Fakkert et al. studied 212 BRCA mutation carriers who underwent prophylactic salpingo-oophorectomy and found no increased fracture risk after five years of follow-up compared with the general population [43]. These discrepancies likely reflect the absence of tumor-related factors in prophylactic cases, in contrast to women with ovarian cancer in whom the malignancy itself directly impairs bone health.

### 4.2. Tumor-Driven Pathways

Ovarian cancer cells exert additional direct effects on bone metabolism, contributing to reduced preoperative BMD in affected women [38]. Tumor cells secrete growth factors that disrupt bone homeostasis, including tumor necrosis factor (TNF), which is elevated in cancer patients [44,45]. Studies in ovarian cancer patients have consistently shown increased RANKL expression, a high RANKL/OPG ratio, and elevated circulating CTX, a marker of bone resorption, reflecting the accelerated bone turnover typical of these patients [46]. Other tumor-derived mediators also promote osteoclast activity. Parathyroid hormone-related peptide (PTHrP), structurally homologous to PTH in its N-terminal domain (1–36), binds to the common PTH/PTHrP receptor to stimulate bone resorption and induce hypercalcaemia [47]. Similarly, prostaglandin E2, found at elevated concentrations in cancer patients, enhances osteoclastic activity and bone resorption [48].

### 4.3. Treatment-Related Mechanisms

Beyond the direct effects of the neoplastic disease itself and the abrupt estrogen deprivation caused by surgical menopause, chemotherapy must also be considered as a relevant factor impacting bone health in women with ovarian cancer.

The most widely used first-line regimen for several years has been paclitaxel 175 mg/m^2^ combined with carboplatin.

Nevertheless, accumulating evidence indicates that chemotherapy further exacerbates bone loss in women with ovarian cancer following salpingo-oophorectomy, and is also associated with concomitant reductions in muscle mass and strength [49]. In a comparative study, Douchi T et al. [50] evaluated women with ovarian cancer treated with surgery followed by cisplatin–adriamycin–cyclophosphamide chemotherapy versus controls who underwent salpingo-oophorectomy for non-malignant conditions. They observed a significantly greater decline in BMD in the chemotherapy group, with mean BMD reduced to 87.4 ± 2.1% after six cycles, compared with 97.6 ± 4% in controls six months post-surgery [50]. Similarly, Hui et al. [51] studied 40 patients with ovarian cancer treated with platinum-based chemotherapy and reported substantial BMD loss: a mean reduction of 10.4% (± 4.06) at the femoral neck and 15.9% (± 5.67) at the lumbar spine (L1–L2) at one year. Bone loss was progressive and persisted beyond the first year, with worsening over time, suggesting that the cytotoxic effect of chemotherapy is directly responsible for skeletal damage, independent of the hypoestrogenic state induced by surgery [51]. By contrast, Nishio et al. reported less pronounced findings in a smaller cohort of 15 women with ovarian cancer treated with carboplatin after salpingo-oophorectomy. In this group, mean BMD decreased to 95.7 ± 3% compared with baseline, a reduction that did not reach statistical significance [52]. Prolonged use of corticosteroids as antiemetics or in recurrence treatment further contributes to osteoporosis.

Collectively, these findings support chemotherapy as an independent driver of skeletal deterioration, with additive effects to surgical menopause and tumor-related pathways.

## 5. Assessment of Bone Health and Management of Osteoporosis in Ovarian Cancer

DXA scanning should be performed to assess bone health, particularly in women with one or more additional risk factors for osteoporosis, such as family history or previous fragility fractures, advanced age, smoking, chronic steroid exposure, concurrent glucocorticoid therapy, reduced mobility, or low BMI.

Although FRAX is commonly used to estimate 10-year fracture risk, it has not been validated in oncology populations, and its predictive accuracy in this context remains uncertain [53]. The initiation of antiresorptive therapy in these women must therefore be considered on an individual basis. Currently, there are no specific guidelines for the management of osteoporosis in patients with ovarian cancer. Nevertheless, clinical practice may be guided by recommendations developed for the management of bone health in breast cancer patients treated with aromatase inhibitors, given the similar pathophysiological mechanisms underlying bone loss in both settings [54,55]. In patients treated with bilateral salpingo-oophorectomy for ovarian cancer, it is recommended to carry out a DXA to estimate BMD and morphometry to evaluate pre-existing vertebral fractures, to obtain an adequate assessment of the bone condition at the onset of surgical menopause [52]. Initiation of medical therapy can be considered in the presence of at least two risk factors: T score < −1.0, age > 65, active smoking or alcohol use, BMI < 20, use of corticosteroids for a period greater than six months, and a family history of fragility fractures. In any case, it should be considered in women with a T score < −2.0. The absence of these criteria, however, does not exclude the need for close follow-up by DXA and reassessment of risk factors every 1 years for these patients, precisely because of the high risk of fracture [54,55]. Once therapy has been started, it is important to verify adequate adherence three months after initiation and to reevaluate patients’ BMD using DXA every two years [52]. A suggested algorithm is proposed in Figure 1.

Pharmacological strategies for bone protection in ovarian cancer focus mainly on antiresorptive agents such as bisphosphonates and denosumab (Table 2).

Anabolic drugs, such as teriparatide or abaloparatide, although capable of increasing bone mass by promoting osteoblastic activity, are not strongly recommended in cancer patients, considering the fears raised by the increased incidence of osteosarcoma recorded in murine models treated with these drugs for a period exceeding 24 months [56]. Bisphosphonates, including alendronate, risedronate, ibandronate, and zoledronate, selectively accumulate in bone by binding to hydroxyapatite crystals and are released during resorption. Once internalized by osteoclasts, they induce apoptosis, thereby suppressing bone turnover and promoting mineralization.

Denosumab, a monoclonal antibody targeting RANKL, effectively mimics the action of OPG and blocks osteoclast activation [53]. Preclinical murine models of ovarian carcinoma provide mechanistic insights. In these models, zoledronate was shown to markedly suppress osteoclastic activity, lowering the RANKL/OPG ratio. Similarly, denosumab treatment preserved trabecular bone completely and partially protected muscle mass and strength [46]. Comparable findings were reported by Essex et al., who demonstrated in murine models that zoledronate, when combined with platinum-based chemotherapy, preserved bone mass and improved both muscle mass (+12%) and strength (+42%) compared with chemotherapy alone [49]. Selective estrogen receptor modulators (SERMs), whose beneficial effects on bone health are well established, may represent a potential therapeutic option in these patients. Since hormone therapy must be used with great caution in this population and the available evidence remains limited, it is noteworthy that several studies have not demonstrated any association between hormone therapy and disease progression in women with non-serous epithelial or germ cell ovarian cancer. In particular, a study involving 94 women with a history of non-serous epithelial ovarian cancer, fallopian tube cancer, or primary peritoneal cancer reported that hormone therapy use was not associated with decreased disease-free or overall survival [57]. Guidozzi and Daponte similarly showed that comparing a group of 130 women with previous epithelial ovarian cancer treated with hormone therapy to a control group, there was no statistically significant difference in disease-free survival between the two groups, with median overall survival of 44 months (95% CI, 10–112 months) and 34 months (95% CI, 8–111 months), respectively [58]. These data appear to be true regardless of the age of the women treated. It must be specified however that the approach is different in patients with serous and granulosa cell tumors, where the use of hormone therapy is not recommended, given the marked hormonal sensitivity of these tumors [30]. However, clinical evidence specific to ovarian cancer remains limited. Therapeutic approaches therefore draw largely on data from general and oncology populations especially with premature menopause. Among available agents, zoledronate is the most potent bisphosphonate, reducing vertebral fracture risk by up to 70% compared with placebo, with proven efficacy also in oncology patients [59]. Moreover, it is widely used for the management of bone metastases and paraneoplastic hypercalcemia, with superior efficacy compared to other antiresorptive drugs, particularly in premenopausal women, where it may be preferred for preserving bone health [55,60]. Denosumab offers comparable protection, as confirmed by a meta-analysis of 11 trials, and maintains a safety profile similar to that of bisphosphonates [61].

To maximize efficacy, antiresorptive therapy should always be combined with adequate calcium (1000 to 1200 mg/d) and vitamin D (at least 800 to 1000 IU/d) supplementation, which modestly reduces fracture incidence but is an essential component of bone-protective strategies in oncology as well as in the general population. In addition to the use of supplements, calcium intake can be ensured by recommending an adequate diet that includes mostly the intake of dairy products, which contain nutrients such as calcium, phosphorus and proteins which are the main nutritional determinants of bone mass accumulation [62]. However, remembering that the main dietary sources of calcium, other than dairy products, include selected low-oxalate vegetables, legumes, nuts, and fortified foods; while for vitamin D, the primary sources are fortified dairy products, fortified foods, and fatty fish [63]. Physicians should encourage patients to practice balance and flexibility exercises, in addition to resistance and strengthening exercises, to reduce the risk of fractures caused by falls [53].

## 6. Conclusions and Future Perspectives

Women with ovarian cancer face a substantially increased risk of osteoporosis and fractures due to the combined effects of surgical menopause, chemotherapy, and tumor-driven bone resorption. Estrogen deficiency accelerates bone remodeling with prevalence of osteoclastic activity, determining rapid bone mass loss, particularly marked in the first year post-surgery. This multifactorial burden translates into higher morbidity, impaired quality of life, and potentially reduced survival. Despite this, bone health is rarely integrated into standard oncological care. Routine DXA assessment, systematic evaluation of fracture risk, and early initiation of antiresorptive strategies, should become core components of ovarian cancer management. These interventions must be complemented by calcium and vitamin D supplementation, physical activity, and fall-prevention strategies (Figure 2).

Closing this gap requires a shift from viewing bone complications as secondary concerns to recognizing them as integral to long-term survivorship. Integrating bone protection into ovarian cancer care is not merely supportive, but a pivotal step towards improving long-term survivorship and quality of life.

## Figures and Tables

**Figure 1 diagnostics-15-02966-f001:**
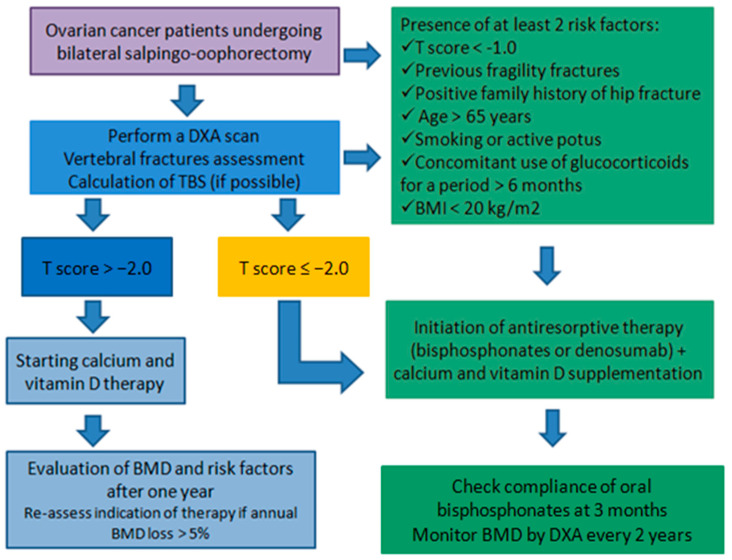
Proposed algorithm for the assessment and management of bone health in ovarian cancer patients undergoing bilateral salpingo-oophorectomy. The flowchart outlines a stepwise approach including DXA scan, vertebral fracture assessment, and trabecular bone score (TBS) evaluation. Based on T-score and clinical risk factors, patients are stratified for calcium and vitamin D supplementation alone or for initiation of antiresorptive therapy (bisphosphonates or denosumab). Regular follow-up includes reassessment of bone mineral density (BMD) and therapy compliance.

**Figure 2 diagnostics-15-02966-f002:**
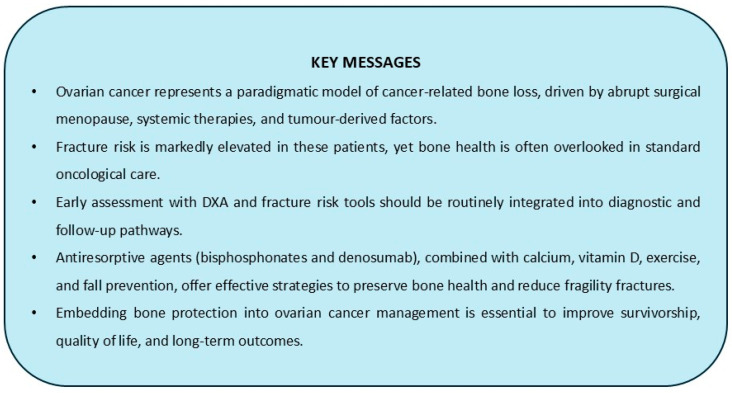
Key messages on bone health in ovarian cancer.

**Table 1 diagnostics-15-02966-t001:** Mechanisms of bone loss in ovarian cancer: biological pathways and clinical consequences.

Mechanism	Biological/Pathophysiological Pathways	Clinical Evidence and Outcomes
Hormonal pathways (estrogen deficiency)	↓ Estrogen → ↓ OPG/↑ RANKL → ↑osteoclastogenesis Inhibition of Fas/FasL-mediated osteoclast apoptosis ↑ TNF, IL-1 ↓ osteoblast survival & mechanotransduction	BMD decline within 1 year; 37–55% osteopenia/osteoporosis; more pronounced <45 yrs; fracture risk debated in prophylactic vs. oncologic cases
Tumor-driven pathways	Tumor-derived cytokines (TNF, IL-1, PTHrP, PGE2) ↑ RANKL/OPG ratio ↑ CTX	Accelerated bone turnover; reduced pre-operative BMD; hypercalcaemia in PTHrP-secreting tumors
Treatment-related mechanisms (chemotherapy)	Platinum/taxane regimens induce direct osteotoxicity and sarcopenia ↑ inflammation, oxidative stress	Additional BMD reduction (−10 to −16% at 1 yr); cumulative chemotherapy cycles predict osteoporosis; persistent loss beyond first year

↓ decrease; ↑ increase.

**Table 2 diagnostics-15-02966-t002:** Management of osteoporosis in women with ovarian cancer: anti-resorptive therapies.

Therapy	Mechanism of Action	Evidence in Ovarian Cancer (Preclinical/Clinical)	Clinical Relevance/Notes
Bisphosphonates (alendronate, risedronate, ibandronate, zoledronate)	Bind to hydroxyapatite → internalised by osteoclasts → induce apoptosis → ↓ bone resorption and turnover	Preclinical: Zoledronate ↓ RANKL/OPG ratio, preserved trabecular bone, improved muscle mass and strength; synergy with chemotherapy. Clinical: Zoledronate ↓ vertebral fracture risk by ~70% in general population; effective in oncology patients, widely used in bone metastases and hypercalcaemia	Zoledronate = most potent; preferred in premenopausal women; long clinical experience
Denosumab	Monoclonal antibody against RANKL → blocks RANK–RANKL pathway (mimics OPG)	Preclinical: preserved trabecular bone and partly muscle mass. Clinical: Meta-analysis of 11 RCTs showed ↓ vertebral fracture risk and maintained BMD with safety comparable to bisphosphonates	Useful alternative when bisphosphonates contraindicated; subcutaneous administration
Calcium + Vitamin D supplementation	Provide substrates for bone mineralization	Clinical: Meta-analyses show modest reduction in hip, vertebral and overall fragility fractures	Recommended in combination with anti-resorptives in both general and oncology populations

→ results in; ↓ decrease.

## Data Availability

The original contributions presented in this study are included in the article. Further inquiries can be directed to the corresponding author.

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
