# Peer review of "Osteoporosis and Fracture Risk in Ovarian Cancer: Beyond the Oncologic Burden"

_diagnostics, 2025, doi:10.3390/diagnostics15232966_

Round 1
Reviewer 1 Report
Comments and Suggestions for Authors
This is an excellent and comprehensive narrative review that addresses an important but often overlooked topic—the relationship between ovarian cancer and skeletal health. The manuscript is clearly written, logically structured, and well referenced. The “triple-hit” concept (surgical menopause, chemotherapy, and tumor-driven bone resorption) provides an effective and original framework for understanding the multifactorial burden on bone metabolism in this population.
The review is scientifically sound, clinically relevant, and integrates mechanistic, diagnostic, and therapeutic aspects in a coherent way. With minor revisions, it will make a valuable contribution to the field of oncology-related bone research.
Specific suggestions:
-
Clarify in the Introduction how this review extends existing knowledge beyond prior reviews on cancer-related osteoporosis (e.g., breast cancer).
-
Expand Section 1.8 to include practical details on DXA monitoring frequency, timing of antiresorptive therapy initiation, and non-pharmacological strategies (exercise, nutrition, fall prevention).
-
Ensure Figures 1 and 2 (algorithm and key messages) are included and clearly labelled.
-
Remove duplicate TNF citations ([41], [42]) and standardize reference formatting (e.g., “[50,52]”).
-
Maintain consistency in spelling (“oestrogen” vs. “estrogen”) and correct minor typographical errors (e.g., “alogorithm” → “algorithm”).
-
The acknowledgment of ChatGPT use is transparent; please ensure the wording complies with Diagnostics policy (e.g., “used for language editing and figure drafting; all intellectual content verified by the authors”).
Author Response
Reviewer 1
This is an excellent and comprehensive narrative review that addresses an important but often overlooked topic—the relationship between ovarian cancer and skeletal health. The manuscript is clearly written, logically structured, and well referenced. The “triple-hit” concept (surgical menopause, chemotherapy, and tumor-driven bone resorption) provides an effective and original framework for understanding the multifactorial burden on bone metabolism in this population.
The review is scientifically sound, clinically relevant, and integrates mechanistic, diagnostic, and therapeutic aspects in a coherent way. With minor revisions, it will make a valuable contribution to the field of oncology-related bone research.
Specific suggestions:
- Clarify in the Introduction how this review extends existing knowledge beyond prior reviews on cancer-related osteoporosis (e.g., breast cancer).
We thank the reviewer for this insightful suggestion. We have revised the Introduction to better define the novelty and scope of our review (lines 65-68 and 73-75).
- Expand Section 1.8 to include practical details on DXA monitoring frequency, timing of antiresorptive therapy initiation, and non-pharmacological strategies (exercise, nutrition, fall prevention).
We appreciate this constructive recommendation. To provide a more comprehensive and cohesive discussion, Sections 1.7 and 1.8 have been combined into a single, expanded section integrating all the requested information on bone health management in ovarian cancer patients, adding the further information you suggested.
- Ensure Figures 1 and 2 (algorithm and key messages) are included and clearly labelled.
We have verified that both Figure 1 (Clinical algorithm) and Figure 2 (Key messages summary) are now consistently formatted, and properly labelled in the revised manuscript and figure legends.
- Remove duplicate TNF citations ([41], [42]) and standardize reference formatting (e.g., “[50,52]”).
We thank the reviewer for noticing this. Duplicate references ([41], [42]) referring to TNF-related studies have been merged, and all reference formatting has been standardized according to the journal’s style (e.g., “[50,52]”). A complete reference cross-check was performed throughout the manuscript.
- Maintain consistency in spelling (“oestrogen” vs. “estrogen”) and correct minor typographical errors (e.g., “alogorithm” → “algorithm”).
All typographical inconsistencies have been corrected. We have standardized spelling to “oestrogen” throughout the text, in line with American English conventions, and corrected minor errors including “alogorithm” → “algorithm”.
- The acknowledgment of ChatGPT use is transparent; please ensure the wording complies with Diagnostics policy (e.g., “used for language editing and figure drafting; all intellectual content verified by the authors”).
We have revised the acknowledgment section accordingly. The new wording is:
“ChatGPT (OpenAI, San Francisco, CA, USA) was used exclusively for language editing and figure drafting. All intellectual content, data interpretation, and conclusions were independently developed and verified by the authors.”
Reviewer 2 Report
Comments and Suggestions for Authors
This manuscript reviews the pathogenesis and management of bone loss in women with ovarian cancer.
- In the Introduction, the statement “The global prevalence of osteoporosis is estimated at around 40%, although this trend varies across different regions of the world [1]” requires clarification. Is the 40% prevalence referring to older adults or the global population as a whole?
- Some information in the review requires citation or correction. For example, the sentence “Clinical risk prediction can be refined using the fracture risk assessment tool (FRAX), which integrates BMD with clinical variables to estimate 10-year fracture probability” should note that the FRAX model can be applied with or without bone mineral density (BMD) measurements.
- Please provide a source for the statement “DXA scanning every 1–2 years and evaluation for pre-existing vertebral fractures are recommended. Additional clinical risk factors should be systematically assessed, and pharmacological therapy should be initiated if at least two such factors are present or if the T-score is ≤ –2.0.” In addition, the caption for Figure 1 should be rewritten for clarity and completeness.
Author Response
This manuscript reviews the pathogenesis and management of bone loss in women with ovarian cancer.
- In the Introduction, the statement “The global prevalence of osteoporosis is estimated at around 40%, although this trend varies across different regions of the world [1]” requires clarification. Is the 40% prevalence referring to older adults or the global population as a whole?
We thank the reviewer for this valuable comment. We have clarified the statement in the revised version of the Introduction, indicating the corrected global prevalence of osteoporosis ranging from 19.8–26.9% in women and 9.6–14.1% in men, with rates as high as 39.5% in certain regions such as Africa.
- Some information in the review requires citation or correction. For example, the sentence “Clinical risk prediction can be refined using the fracture risk assessment tool (FRAX), which integrates BMD with clinical variables to estimate 10-year fracture probability” should note that the FRAX model can be applied with or without bone mineral density (BMD) measurements.
We appreciate the reviewer’s observation. We have revised the sentence to clarify this point and included the appropriate reference (Kanis JA et al. Osteoporosis Int 2008).
- Please provide a source for the statement “DXA scanning every 1–2 years and evaluation for pre-existing vertebral fractures are recommended. Additional clinical risk factors should be systematically assessed, and pharmacological therapy should be initiated if at least two such factors are present or if the T-score is ≤ –2.0.” In addition, the caption for Figure 1 should be rewritten for clarity and completeness.
We thank the reviewer for this helpful suggestion. We have now cited these references in the revised manuscript and rephrased the paragraph for precision.
The caption for Figure 1 has also been revised for clarity as follows: Figure 1. Proposed algorithm for the assessment and management of bone health in ovarian cancer patients undergoing bilateral salpingo-oophorectomy. The flowchart outlines a stepwise approach including DXA scan, vertebral fracture assessment, and trabecular bone score (TBS) evaluation. Based on T-score and clinical risk factors, patients are stratified for calcium and vitamin D supplementation alone or for initiation of antiresorptive therapy (bisphosphonates or denosumab). Regular follow-up includes reassessment of bone mineral density (BMD) and therapy compliance.